# Siderophore (from *Synechococcus* sp. PCC 7002)-Chelated Iron Promotes Iron Uptake in Caco-2 Cells and Ameliorates Iron Deficiency in Rats

**DOI:** 10.3390/md17120709

**Published:** 2019-12-16

**Authors:** Xue Feng, Suisui Jiang, Fan Zhang, Runfang Wang, Yuanhui Zhao, Mingyong Zeng

**Affiliations:** College of Food Science and Engineering, Ocean University of China, Qingdao, Shandong 266003, China

**Keywords:** *Synechococcus* sp. PCC 7002, siderophores, Caco-2 cell, iron-deficiency anemia, anemia intervention

## Abstract

Siderophores are iron chelators with low molecular weight secreted by microorganisms. Siderophores have the potential to become natural iron fortifiers. To explore the feasibility of the application of *Synechococcus* sp. PCC7002-derived siderophores as iron fortifiers, *Synechococcus* sp. PCC7002, as a carrier, was fermented to produce siderophores. The absorption mechanism and anemia intervention effect of siderophores-chelated iron (SCI) were studied through the polarized Caco-2 Cell monolayers and the rat model of iron-deficiency anemia, respectively. The results indicated that siderophores (from *Synechococcus* sp. PCC7002) had an enhancing effect on iron absorption in polarized Caco-2 cell monolayers. The main absorption site of SCI was duodenum with pH 5.5, and the absorption methods included endocytosis and DMT1, with endocytosis being dominant. The effect of sodium phytate on SCI was less than that of ferrous sulfate. Therefore, SCI could resist inhibitory iron absorption factors in polarized Caco-2 cell monolayers. SCI showed significantly higher relative bioavailability (133.58 ± 15.42%) than ferrous sulfate (100 ± 14.84%) and ferric citrate (66.34 ± 8.715%) in the rat model. Food intake, hemoglobin concentration, and hematocrit and serum iron concentration of rats improved significantly after Fe-repletion. Overall, this study indicated that siderophores derived from *Synechococcus* sp. PCC7002 could be an effective and feasible iron nutritive fortifier.

## 1. Introduction

Iron plays an important role in maintaining normal human metabolism and is essential for most organisms [1]. However, iron deficiency is a ubiquitous micronutrient deficiency worldwide. Iron-deficiency anemia (IDA) caused by iron deficiency could lead to multiple pathologies, especially delays in development and behavior among children [2,3,4]. Over two billion people worldwide suffer from this problem, according to the FAO (Food and Agriculture Organization) [2,5].

Iron deficiency and IDA arise when requirements cannot be met by dietary iron, particularly in developing countries, primarily due to insufficient iron content in food and low iron absorption efficiency. This is because iron digestion and absorption are often inhibited by the various dietary components, such as oxalates and phytate. Therefore, food fortification and supplementation—with ferrous sulfate and ferric sodium EDTA, for example—are commonly used to combat IDA [6]. However, ferrous sulfate could affect appetite and digestion due to its side effects on the stomach, and ferric sodium EDTA is a synthetic chelating agent, the safety of which is a concern for the public, resulting in lower daily intakes [7]. Therefore, improving the intake and bioavailability of dietary iron is considered to be an effective way to combat IDA [8,9].

Siderophores are low-molecular-weight compounds secreted by microbes, several fungi, and plants under limited iron condition [10]. They could facilitate acquisition of iron and counter the iron deficiency by high-affinity iron (III) ligands [10]. Compared with other metal elements, iron is preferentially chelated by siderophores [11]. Moreover, only ferric iron could be chelated by siderophores. When ferric iron is reduced to ferrous iron, it will be released by siderophores. It is an important strategy for microbes to combat the iron stress environment [10,12]. The role of siderophores in microbial physiological activities is to dissolve ferric iron and transport it into the cytoplasm, so siderophores have the potential to become a nutritional fortifier [13]. The hypothesis was partly verified in the plant nutrition field.

Siderophores have two major structural features, α-Hydroxy-carboxylate acid (α-hydroxy aspartic acid and citric acid) and amphipathy in the form of polar head and hydrophobic tail. The majority of marine siderophores have both distinctive structural features [10]. Accordingly, they both contain amphipathy and photochemistry in Fe(III) complexes. Siderophores could anchor a certain gradient outside the cell membrane, depending on membrane affinity and length, thereby enhancing the ability to capture iron from seawater [12,14].

*Synechococcus* sp. PCC7002 is an elucidated unicellular marine cyanobacterium and could produce siderophores of multiple types. After optimizing the culture conditions, *Synechococcus* sp. PCC7002 can grow at a multiple rate within three hours. It could also produce more ferritin under iron-limited conditions. In particular, siderophores with the structure of α-Hydroxy-carboxylate, which were separated by Martinez, have different fatty acid chains and hang outside the cell membrane to capture ferric iron [15]. Therefore, the microalgae are a suitable carrier strain for producing siderophores. Our previous study identified the culture conditions of *Synechococcus* sp. PCC7002; therefore, it has the foundation to become an iron nutrition enhancer.

This paper aimed to study the intestinal absorption effect of SCI by the polarized Caco-2 Cell monolayers and verify the anemia intervention effect of SCI through the iron-deficiency anemia rat model. By analyzing the iron intestinal absorption effect and bioavailability of SCI, the feasibility of SCI to become an iron nutrition enhancer was explored.

## 2. Results

### 2.1. Analysis of Siderophore Absorption Site in Polarized Human Intestinal Epithelial (Caco-2) Cells

As shown in Figure 1a, the fluorescence intensity exhibited a sharp decrease over culture time, which indicated that a high SFE-Fe absorption appeared in polarized Caco-2 cell monolayers. The absorption at pH 5.5 was higher than that at pH 7.0, which showed that acidic the environment was more suitable for SFE-Fe absorption than the neutral one. This result could be explained by the fact that exogenous iron entered the acidic duodenum first, followed by the jejunum and then the other parts of the intestine during iron digestion and absorption [16,17]. Therefore, consistent with other forms of iron, SFE-Fe was mainly absorbed in the duodenum. Because intestinal iron absorption primarily occurred in the duodenum close to the stomach (when fasting, pH = 6.1; when eating, pH = 5.5) [18]. For clear results, only the result of pH 5.5 was shown in the following experiments.

### 2.2. The Effect of Different Extraction Methods on Iron Absorption

As shown in Figure 1b, when SFE-Fe, SCE-Fe, and SFP-Fe (from different extraction methods) were added, respectively, the trend of fluorescence intensity was consistent. Therefore, different extraction methods and purity had no difference in regard to iron absorption. The following experiments showed only the result of SFE-Fe.

### 2.3. Analysis of Siderophore Absorption Methods in Polarized Caco-2 Cell Monolayers

Most of the iron was absorbed through DMT1 (Ferrous iron) or endocytosis (ferric oxyhydroxide nanoparticles) [8,19,20]. BPDS is a powerful complexing agent of ferrous ion. Excessive BPDS could block the absorption and transport pathway of divalent iron mediated by DMT1 through complexing all ferrous ions in solution. Ferrous sulfate, a ferrous salt, was absorbed only through the DMT1 pathway. Therefore, as shown in Figure 2a, it was fully blocked by BPDS. However, SFE-Fe was almost unaffected by BPDS. Compared to SFE-Fe only, the absorption of SFE-Fe with BPDS had no significant difference. Compared to SFE-Fe only, the fluorescence intensity of SFE-Fe with BPDS significantly decreased, but by less than 5%. The results suggested that SFE-Fe was absorbed primarily by endocytosis directly, and a small portion of the iron was absorbed through the DMT1 pathway.

### 2.4. Analysis of Reduction and Absorption of SFE-Fe in Polarized Caco-2 Cell Monolayers

The reduced iron content of SFE-Fe by Dcytb could be measured according to the determination of ferrous iron, which combined with BPDS and could be analyzed through colorimetry thereafter [21]. The reduced iron content observed during the iron-absorption experiment was measured (Figure 2b) when iron was chelated by SFE, sodium phytate, EDTA, and DFB, respectively. SFE-Fe was more easily reduced than EDTA (common chelating agent), DFB (a kind of siderophores), and sodium phytate (*p* < 0.01). The result indicated that the reduction of SFE-Fe was inclined to Dcypt, rather than to others. SFE-Fe was easily reduced to ferrous iron natively and subsequently absorbed due to the unique existing structure of citric acid and hydroxamic acid. The fact that the absorption of SFE-Fe was blocked by BPDS also partly supported this possibility (Figure 2a).

### 2.5. Effect of Sodium Phytate on the Absorption of Siderophores-Chelated Iron Intestinal Cells

Iron was easily combined with some plant food ingredients induced to the formation of precipitate, which could hinder iron absorption in intestinal cells [22]. Some iron fortifiers, especially ferrous salt, showed a good effect on iron absorption in vitro [23,24]. However, these fortifiers were subject to precipitation by organic acids in the intestines, and the bioavailability was sharply decreased [25]. As shown in Figure 3a, a significant difference was found between SFE-Fe and ferrous sulfate with the addition of equal concentrations of sodium phytate. Sodium phytate had six ligands. It could chelate with ferrous iron or higher metal ions to form precipitation rapidly under acidic conditions. Therefore, when sodium phytate was added, the absorption of ferrous sulfate was seriously inhibited. A significant reduction of SFE-Fe absorption was found with sodium phytate added. Therefore, sodium phytate did not have a significant inhibition effect on SFE-Fe compared to ferrous sulfate.

### 2.6. Effect of SFE-Fe on the Metabolic Balance in Intestinal Cells

To investigate the effect of SFE-Fe on metabolic balance for intestinal cells, three different concentration (20, 40, and 60 μM) of SFE-Fe were added in Caco-2 cells, and the fluorescence intensity change of calcein is shown in Figure 3b. A sharp increase of LIP iron appeared in the first 25 min, and then the increase rate was greatly reduced until the calcein fluorescence value stabilized. Moreover, no significant difference in the final stable fluorescence intensity was found during these time periods. The result indicated that the absorption of SFE-Fe was adjusted by intestinal iron metabolism balance mechanism, which could not be destroyed by the intake of SFE-Fe.

### 2.7. Measurement of Diet Intake on the Growth and Development of Rats

Iron deficiency could cause low hemoglobin concentration and deficiency of iron enzyme, which initiates systemic dysfunction and abnormal macrophage secretion, further damaging immune function [8,26,27]. The rat body weight and diet intake of different groups are shown in Table 1. The diet intake of ferrous sulfate was significantly lower than SFE-Fe. It might because that inorganic salt, ferrous sulfate had side effects on the stomach and affected appetite and digestion [9]. The weight gain of SFE-Fe was significantly higher than ferrous sulfate and ferric citrate groups, which suggested that SFE-Fe had considerable advantages in promoting appetite and growth in rats. There was no significant difference between ferrous sulfate and ferric citrate regarding weight after the Fe-repletion period. Apparently, due to the side effect on the intestines, ferrous sulfate influenced the body weight gain of rats and showed no difference with ferrous citrate (trivalent iron), which was also harmless to the intestines.

### 2.8. Measurement of Hb, Hct, and Serum Iron

The change in Hb, Hct, and serum iron content of different groups was shown in Table 2. For Hb content, no significant difference was found between the control group and the Fe-deficient group. The level of Hb of SFE-Fe group was significantly higher than the ferrous sulfate group, and the Hb content of ferrous sulfate group was also higher than that of the ferric citrate group. Hct was used to reflect blood viscidity, the level of which was very low after the Fe-deficient period, which was consistent with blood thinning [28]. The Hct level of the Fe-deficient group showed significant reduction after the Fe-depletion period, while the Hct level of groups with the addition of exogenous iron increased sharply. No difference of Hct content was found between SFE-Fe and ferrous sulfate. The result suggested that, when enough external iron was supplemented, ill status caused by low red blood cell volume could receive priority relief. Serum iron refers to iron bound to transferrin, the low level of which could indicate successful modeling. After Fe-repletion, the serum iron content of every group increased with different degrees, especially in the SFE-Fe group, and the ferrous sulfate group.

### 2.9. Relative Bioavailability of SCE-Fe

Iron mainly exists in hemoglobin (65%), macrophages, and liver in mammals [29]. As shown in Figure 4a, though exogenous iron was added to supplement iron after Fe-depletion, liver iron content of the Fe-deficient group was significantly lower than that of the control group. There was no significant difference between the exogenous iron groups, and it also showed no significant difference with the negative group.

The relative bioavailability of various groups is shown in Figure 4b. A highly significant difference is shown between them. The bioavailability of SCE-Fe increased notably compared with ferric citrate alone, which was also higher than the value obtained with ferrous sulfate, the positive control. The result suggested that SCF-Fe could promote intestinal absorption of trivalent iron considerably and had strong anemia intervention ability.

## 3. Discussion

As a marine siderophore, SFE is specific for the chelation of iron and binds only to ferric ions [11]. SFE has the structure of α-Hydroxy-carboxylate acid (α-hydroxy aspartic acid and citric acid) and amphipathy in the form of polar head and hydrophobic tail [10]. Both contain amphipathy and photochemistry in Fe(III) complexes. Therefore, when algae cells absorb and bind iron, the trivalent iron is reduced to ferrous iron, released due to photodecomposition, and subsequently absorbed via ferrous iron transport [12]. Therefore, as shown in Figure 2a,b, the absorption of SFE-Fe was influenced less by BPDS. Moreover, siderophores, a powerful biological chelating agent, have the potential to be absorbed directly by endocytosis in the intestinal cells, similar to small peptide-chelated iron and ferritin-chelated iron [20]. Chen et al. [7] reported the absorption model of siderophore-chelated iron, which was absorbed by endocytosis when it helped soybeans to resist iron stress.

The photolysis reaction occurring in the unique citric acid–Fe complex of marine siderophores suggested the reducibility of it [12]. Additionally, some siderophore-chelated iron (III) was reduced to ferrous iron by Dcytb reductase and absorbed later through DMT1 on the intestinal epithelial cell surface [19]. We assumed that some trivalent iron chelated by siderophores was reduced to ferrous iron by Dcytd. With ferrous iron released, sodium phytate could combine with it to form a precipitate. As shown in Figure 3a, the absorption of SFE was hardly affected by sodium phytate. This result indicated that siderophores are a powerful biological chelating agent. When combined with trivalent iron, siderophores were primarily absorbed by endocytosis, which could not be depleted by sodium phytate. Therefore, siderophores had a strong ability to resist inhibiting iron absorption.

With external iron added, the serum index level of model groups was improved significantly, and no significant difference was found between the SFE-Fe and ferrous sulfate groups. Therefore, the effect of iron supplementation needed to be measured by calculating the relative bioavailability of iron. As shown in Figure 2a the absorption rate of ferrous sulfate was higher than that of SCE-Fe in intestinal cells, while the relative bioavailability was inverted in animal experiments (Figure 4b). It might because that ferrous iron was prone to be oxidized to trivalent iron by oxidizing components from foods during digestion [30], which induced a reduction in bioavailability. On the other hand, ferrous sulfate was prone to be affected by inhibitory components, such as oxalic acid, phytic acid, and polyphenol [31], while SCE-Fe was less affected. The result was consistent with the conclusion that SFE was almost unaffected by sodium phytate in polarized Caco-2 Cell monolayers (Figure 3a). Therefore, the absorption of SCE-Fe was hardly affected by food ingredients.

## 4. Methods and Materials

### 4.1. Materials

*Synechococcus* sp. PCC7002 was cultivated in the laboratory with Medium, at 30 °C, under a light intensity of 140 μmol m^−2^ s^−1^, in 10 L glass photobioreactors (Figure 5). All Wistar rats were obtained from Shandong Lukang Pharmaceutical Group Co., Ltd. (Shandong, China). Pelletized purified AIN-93G-based diets were obtained from TROPHIC Animal Feed High-Tech Co., Ltd. (Nantong, China). Calcein acetoxymethylester (calcein-AM), cell culture medium (DMEM), and fetal bovine serum were purchased from Gibco (Grand Island, NY, USA). XAD-2 resin, deferoxamine mesylate, sodium phytate, and bathophenanthrolinedisulfonic acid (BPDS) were obtained from Sigma-Aldrich Co. (St. Louis, MO, USA). All other reagents were analytical grade or better.

### 4.2. Preparation of Synechococcus sp. PCC7002 Siderophores

*Synechococcus* sp. PCC7002 was harvested and centrifuged at 6000× *g* for 15 min, to collect supernatant through a hypervelocity-speed refrigerated centrifuge (GL-21M, Xiangyi, China). After concentration, the supernatant was desalted to obtain siderophore fermentation products (SFP). For obtaining siderophore crude extract (SCE), SFP was purified through the general XARD-2 resin method as follows [32], exhibiting a relatively low purity: The pH of the SFP was adjusted to 3 in order to release iron from the siderophore. It mixed well with the pretreated XARD-2 resin and was stirred with for 6 h. After loading onto the column, it was eluted with Milli-Q water (Millipore, USA) and with 4 column volumes of ultrapure water, and then eluted with 2 to 4 column volumes of 100% methanol. The obtained liquid sample was rotary-screwed by a rotary evaporator, to remove the methanol, and the mixture was continuously steamed to dryness and then added with water (repeated three times). The final aqueous solution was freeze-dried to obtain a powdery sample. The purity is relatively low, called siderophore crude extraction (hereinafter referred to as SCE). After XARD-2 resin loading onto the column, the SFP with pH 3 was eluted at a flow rate of <0.1 L/min, and then it was eluted with 4 column volumes of ultrapure water, 50% methanol, and 100% methanol, respectively. Finally, the sample was washed and concentrated according to the above methods. The final power had a higher purity (SFE). The siderophore extracts were arranged in the following order of purity: SFE > SCE > SFP. The samples (obtained by the three ways) of chelated iron all belong to SCI. The siderophore extracts obtained in three different ways are abbreviated in this study.

Siderophores-chelated iron and other samples: FeSO_4_ (20 uM); the SFE, SCE, and SFP were dissolved in distilled water, adding the same volume of FeCl_3_ • 6H_2_O. After 6 h of reaction at pH = 6.0 ± 0.5, we obtained SFE-Fe, SCE-Fe, and SFP-Fe, and the concentration of FeCl_3_ • 6H_2_O was 40 uM (ICC = 41.88 + 1.70 uM). Before the samples were added into Caco-2 cells, the samples were diluted with 2 times Tyrode solution (137 mM of NaCl, 2.7 mM of KCl, 1 mM of MgCl2, 1.8 mM of CaCl2, and 5.5 mM of D-glucose). The same method was used for the chelate solution of iron with EDTA, DFB, and sodium phytate, respectively.

### 4.3. Construction of the Polarized Caco-2 Cell Monolayers

Caco-2 cell, obtained from Cell Bank of the Chinese Academy of Sciences (Shanghai, China), was used as a cellular model to study human intestinal absorption. The differentiated cells were obtained according to the method of WU et al. [8], known as the in vitro intestinal model.

Caco-2 cells were seeded at a density of 5 × 104 cells/cm^2^, in collagen-treated 24-well plates. The cells were cultured in medium containing high-glucose Dulbecco’s modified Eagle medium with 10% fetal bovine serum, 2 mM of L-glutamine, and 1% penicillin-streptomycin, at 37 °C, in a CO_2_ incubator (Heraeus, Germany). The medium was renewed every 2 days. After growth of cell fusion (about 2–3 days), the cells were allowed to differentiate in completed DMES medium for 12 days, to differentiate completely.

The cells were incubated in iron-starvation medium for 24 h, with 4 mg/L of hydrocortisone, 5 mg/L of insulin, 5 μg/L of selenium, 34 μg/L of triiodothyronine, and 20 μg/L of epidermal growth factor for the following experiments.

### 4.4. Intestinal Iron Absorption in the Polarized Caco-2 Cell Monolayers

After absorption by intestinal epithelia, exogenous iron was transformed into a part of the labile iron pool (LIP), rather than being stored in the form of transferrin directly or being synthesized iron-containing enzymes [33]. Therefore, cell iron absorption could be reflected by the iron content of LIP, which negatively correlated with the fluorescence intensity of calcein [34]. Therefore, the absorption of iron content was measured with a calcein kit.

The SFE-Fe was added to culture medium of polarized Caco-2 cell monolayers under different pH conditions (pH = 5.5 and pH = 7.0), to obtain the optimum pH [35]. The absorption mechanism of siderophore was studied by adding BPDS under optimum pH [8,36].

### 4.5. Animal Experiments

All animal handling was conducted according to the principles and guidelines outlined in the National Institutes of Health (NIH) Guide for the Care and Use of Laboratory Animals and approved by the Ethical Committee of Animal Care and Use at Ocean University of China (Permit 20001013).

Thirty-two male Wistar male and their lactating mothers were fed Fe-deficient diets at 16 days of age, and young rats were fed normal diets simultaneously. Another 8 pups and their lactating mothers were fed normally. When the pups were 22 days old, they were all weaned and separated from the female rats. Thirty-two rats fed Fe-deficient diets continued with the same treatment, and the other eight normal pups continued to be fed a normal diet for 32 days. The Hb concentration of rats was measured after 32 days for the Fe-deficient group by tail-vein bleeding [2,8].

The Fe-deficient Wistar rats were randomly assigned into FeSO_4_, SFE-Fe, ferric citrate, and Fe-deficient groups. Another eight normal pus were still given a free diet as normal groups. Rats from various groups received the corresponding treatment for 16 days, while Fe-deficient groups were treated with a Fe-deficient diet (Figure 6). After Fe-repletion, the Hb concentration of rats were measured by tail-vein bleeding after fasting for 16 h. The rats were anesthetized, and blood was drawn from their abdominal aortas. The livers of the rats were removed, frozen with liquid nitrogen, and stored at −80 °C.

### 4.6. Hb, Hct, and Serum Iron Measurements

Hb and Hct were measured through an automated hematology analyzer (Affiliated Hospital of Qingdao University, Qingdao, China). The content of serum iron was measured using a serum iron assay kit (Nanjingjiancheng, Wuhan, China) through an Enzyme mark instrument (Synergy H4, Bio-Tek, USA).

### 4.7. Relative Bioavailability of SCE-FE

After drying to constant weight in the oven, at 90 °C, the liver was weighed and digested to measure iron content by using UV-2550 visible spectrophotometry (SHIMADZU, Japan).

The relative bioavailability of SCE-FE was measured according to the method of Hilty [37].

The Hb iron pool (mg), assuming a total rat blood volume of 6.7% body weight (BW) and an iron content in Hb of 0.335%, was calculated as follows:

Hb iron pool (mg) = BW (kg) × 0.067 × Hb concentration (g/L) × 3.35.

Hb regeneration efficiency (%) = [final Hb iron pool (mg) - initial Hb iron pool (mg)]/total iron consumed (mg) × 100.

The relative biological value was calculated from the Hb regeneration efficiencies:

The relative biological value (%) = Hb regeneration efficiency of each animal/Hb regeneration efficiency of the FeSO_4_ group × 100.

### 4.8. Statistical Analysis

The experimental data were based on triplicate analysis. The statistical analysis was performed by using SPSS 24.0 software. Data were expressed as the mean ± standard deviation. Mean values were obtained by statistical analysis of variance (one-way ANOVA). Significant differences in treatment means were identified at the level of *p* < 0.05.

## 5. Conclusions

This study revealed that SCI was absorbed in the intestine and not affected by sodium phytate, which indicated its good anti-interference ability to food ingredients during iron absorption. SCI exhibited a significant improvement in iron-deficiency anemia, and the bioavailability is higher than that of ferrous sulfate and ferric citrate. Therefore, siderophores derived from *Synechococcus* sp. PCC7002 could be considered a promising and feasible iron nutritive fortifier supplementation.

## Figures and Tables

**Figure 1 marinedrugs-17-00709-f001:**
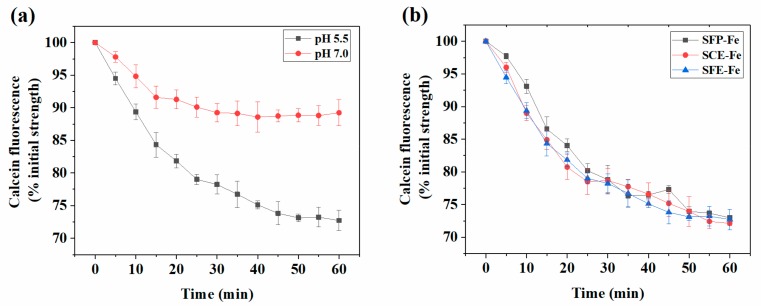
(**a**) Iron intake content of SFE-Fe in polarized Caco-2 cell monolayers with different pH. (**b**) Iron uptake content of SFE-Fe, SCE-Fe, and SFP-Fe in polarized Caco-2 cell monolayers. Values are expressed as means ± standard deviations (n = 3).

**Figure 2 marinedrugs-17-00709-f002:**
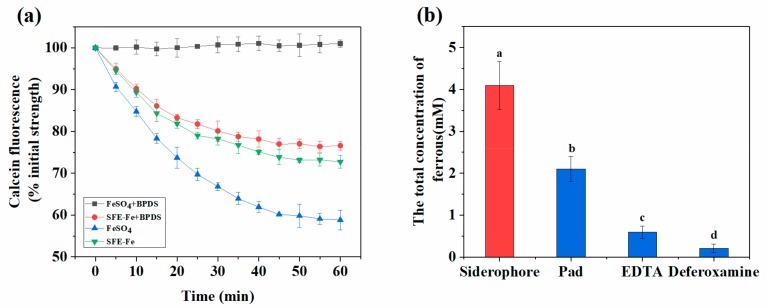
(**a**) Iron intake content of SFE-Fe in polarized Caco-2 cell monolayers when BPDS adding. (**b**) The ferrous concentrations of different sample when incubated for 2 h in polarized Caco-2 cell monolayers (Pad: sodium phytate). Values are expressed as means ± standard deviations (n = 3). Bars with different letters are significantly different (*p* < 0.05).

**Figure 3 marinedrugs-17-00709-f003:**
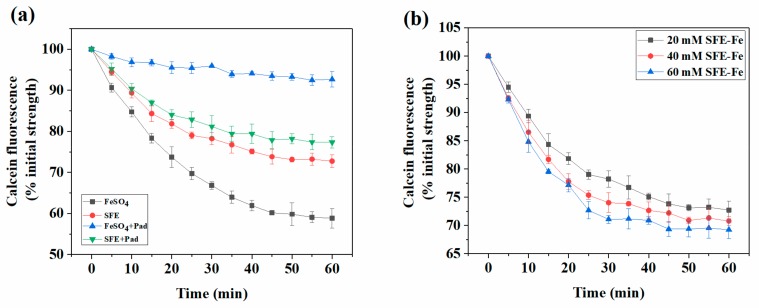
(**a**) The effect of sodium phytate on iron uptake of SFE-Fe in polarized Caco-2 cell monolayers (Pad: sodium phytate;). (**b**) Iron uptake content of SFE-Fe with different concentrations in polarized Caco-2 cell monolayers. Values are expressed as means ± standard deviations (n = 3).

**Figure 4 marinedrugs-17-00709-f004:**
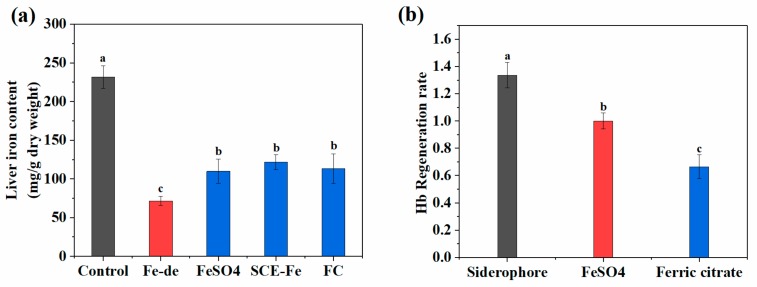
(**a**) Liver iron concentrations of different groups at the end of iron repletion period (FC-de: Fe-deficient, FC: ferric citrate). (**b**) Relative biological values of iron of different iron supplements. Values are expressed as means ± standard deviations (n = 8). Bars with different letters are significantly different (*p* < 0.05).

**Figure 5 marinedrugs-17-00709-f005:**
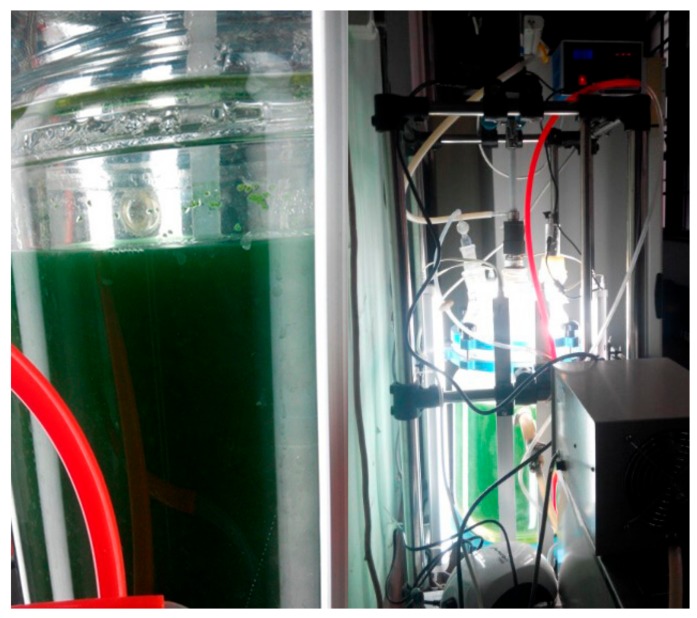
Synechococcus sp. PCC 7002 cultivation in 10 L glass photobioreactors.

**Figure 6 marinedrugs-17-00709-f006:**
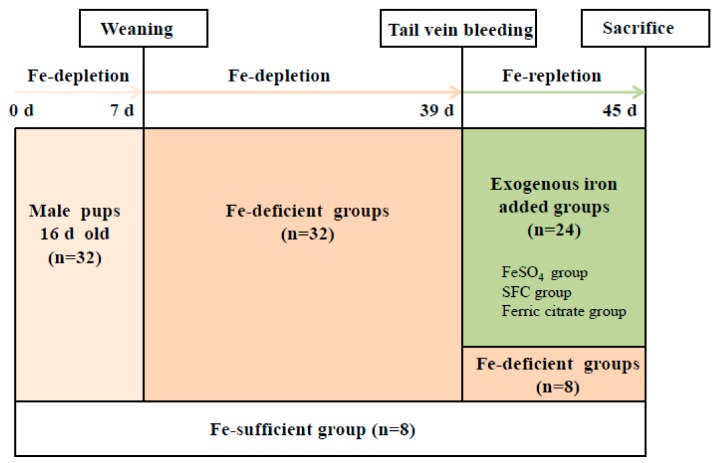
Schematic representation of the plan of animal experiment (the variable n represents the animal number).

**Table 1 marinedrugs-17-00709-t001:** The weight changes and diet intakes of rates at the end of iron repletion period. Values are expressed as means ± standard deviations (n = 8). Different lowercase letters in the same column indicate significant differences between groups and intragroup, respectively; (*p* < 0.05).

Group	Iron Repletion				
	Control	Fe-Deficient	FeSO_4_	SFC-Fe	Ferric Citrate
N	8	8	8	8	8
Diet iron (mg/Kg)	36.7 ± 0.12	13.08 ± 1.34	29.11 ± 0.9	29.56 ± 1.3	33.32 ± 0.92
Diet intake (g)	348.64	247.36	286.77	316.01	259.78
Initial BW (g)	254.5 ± 25.66 ^a^	170.13 ± 28.74 ^b^	170.33 ± 31.76 ^b^	171.25 ± 18.99 ^b^	170.13 ± 19.08 ^b^
Final BW (g)	356.5 ± 32.93 ^a^	205.38 ± 31.15 ^b^	245.67 ± 45.46 ^c^	292.1 ± 34.91 ^d^	231.63 ± 26.94 ^c^
BW changes (g)	102 ± 9.77 ^a^	30.25 ± 12.29 ^b^	75.33 ± 14.83 ^c^	120.75 ± 17.87 ^d^	61.5 ± 9.57 ^e^

BW: body weight.

**Table 2 marinedrugs-17-00709-t002:** The blood indexes of rats at the end of iron repletion period. Values are expressed as means ± standard deviations (n = 8). Different lowercase letters in the same column indicate significant between groups and intragroup differences, respectively; (*p* < 0.05).

Group	Iron Repletion				
	Control	Fe-Deficient	FeSO_4_	SFC-Fe	Ferric Citrate
Initial Hb (g/L)	162.25 ± 20.50 ^a^	74.5 ± 13.48 ^b^	73.11 ± 14.24 ^b^	72.75 ± 12.73 ^b^	71.75 ± 15.95 ^b^
Final Hb (g/L)	166.25 ± 22.55 ^a^	71.5 ± 20.97 ^b^	132.44 ± 16.99 ^c^	145.75 ± 16.51 ^c^	113 ± 21.82 ^d^
Hb change (g/L)	4 ± 3.74 ^a^	−3 ± 9.14 ^a^	59.33 ± 5.27 ^b^	73 ± 5.89 ^c^	41.25 ± 7.14 ^d^
Initial Hct(%)	45.65 ± 4.09 ^a^	15.9 ± 4.43 ^b^	14 ± 3.32 ^b^	14.48 ± 3.55 ^b^	15.35 ± 5.22 ^b^
Final Hct (%)	48.13 ± 6.88 ^a^	10.5 ± 5.55 ^b^	36.78 ± 4.27 ^c^	39.25 ± 9.56 ^c^	28.15 ± 9.61 ^d^
Initial serum iron (μg/g)	35.46 ± 5.14 ^a^	10.37 ± 2.21 ^b^	10.47 ± 3.57 ^b^	10.6 ± 2.74 ^b^	10.6 ± 1.99 ^b^
Final serum iron (μg/g)	35.57 ± 7.62 ^a^	8.72 ± 3.32 ^b^	28.59 ± 10.76 ^c^	31.69 ± 8.85 ^c^	22.26 ± 6.11 ^d^

Hb: hemoglobin; Hct: hematocrit.

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
