# Peer review of "Siderophore (from Synechococcus sp. PCC 7002)-Chelated Iron Promotes Iron Uptake in Caco-2 Cells and Ameliorates Iron Deficiency in Rats"

_marinedrugs, 2019, doi:10.3390/md17120709_

Round 1

Reviewer 1 Report

The authors made all the corrections suggested by the reviewers. I recommend this paper for publication

Author Response

Dear reviewers,
We would like to thank you and the reviewers for thoughtful critiques of our manuscript. We have adopted most of the suggestions, and think that the manuscript has been greatly improved by these revisions. We hope that you will now find it suitable for publication in marine drugs. Our point-by-point responses to the comments are detailed on the attched .

Sincerely,

Reviewer 2 Report

The authors have satisfactorily addressed the questions raised by the reviewers.

Author Response

Dear reviewers,
We would like to thank you and the reviewers for thoughtful critiques of our manuscript. We have adopted most of the suggestions, and think that the manuscript has been greatly improved by these revisions. We hope that you will now find it suitable for publication in marine drugs. Our point-by-point responses to the comments are detailed on the attched .

Sincerely,

This manuscript is a resubmission of an earlier submission. The following is a list of the peer review reports and author responses from that submission.

Round 1

Reviewer 1 Report

This is an interesting study however, there are several grammatical and stylistically errors, which need to be corrected. I recommend the article to be checked by a native speaker.

Did you measure the levels of ferritin, transferrine etc.? Those are essential parameters of iron storage levels. This matter should be discussed. Please, divide the Results and discussion section as the journal requires. Describe more exact the process of tissue harvesting. Did you use purified water to prepare your samples ? Did you use steel equipment (manganese may be a parto of alloy) The reference list is outdated. There are several more recent studies discussing this topic.

Grochowski, C., Blicharska, E., Baj, J., Mierzwińska, A., Brzozowska, K., Forma, A., & Maciejewski, R. (2019). Serum iron, Magnesium, Copper, and Manganese Levels in Alcoholism: A Systematic Review. Molecules, 24(7), 1361. doi:10.3390/molecules24071361

Reviewer 2 Report

The authors have identified a marine siderophore which can potentially be used to fortify iron and increase its absorption. Although the research presented has some evidence to show but the conclusions drawn by the authors are sometimes exaggerations and should be altered. I have a few points that could be addressed to make this better.

In 2.1 the authors show the experimental data from Caco cell absorption but they conclude that SFE-Fe was mainly absorbed in the duodenum? This is unclear.

In 2.2 there appears to be a typo with BDPS and DPBS.

In 2.5 On the one hand, the excessive 143 iron in LIP could be transported outside the cells; on the other hand, a series of changes of membrane 144 permeability and Dcytb occurred in the intestinal cell surface, which could reduce the continuous 145 absorption of iron [22].

While the table suggests that the experimental groups had 8 mice each the legend has only 3 mice to do the statistical analysis? Or were these experiments repeated 3 times on 8 mice each, this is unclear in all mouse experiments.

In the table what does after model and after recovery refer to?

Statements like this are vague and it is unclear whther the authors are referring to a previous study or their own data, which is not in the paper “On the one hand, ferrous iron was prone to be oxidized to trivalent iron by oxidizing components from foods during digestion, which induced a reduction in  bioavailability. On the other hand, ferrous sulfate was prone to be affected by inhibitory components, such as oxalic acid, phytic acid, and polyphenol, while SFE-Fe was less affected”

In the Caco cell model section in methods what did the authors use the differentiated cells for? The methods suggest they isolated cells from the rats? Where are those experiments?

I wouldn’t classify Caco-2 cell iron absorption as in vivo intestinal absorption, the intestine has different regions which absorb differently, this can simply be called invivo iron absorption studies in Caco2 cells.

The kit used to measure serum iron content was a hepcidin ELISA kit? (line 252)

The experimental groups are unclear from the way the results have been presented, the authors claim that treatment with SCI-Fe treated iron defieciency anemia- or was at least beneficial but it is unclear from the way the data is presented.

Did the authors look at the effects of this siderophore on iron homeostasis in the rats? It wil be important to see how this treatment affects the expression of genes involved in iron homeostasis in the livers of these rats. This can be done by simple RNA expression experiments using real time PCR.

 There are multiple instances of the authors suggesting the potential mechanism of absorption through DMT1 but it is unclear from the evidence provided. I would suggest doing the experiments done in Figure 1 in CaCO2 cells with a DMT1 knockdown to provide conclusive evidence of DMT1 involvement.